# LC-MS-Based Metabolomics for the Chemosystematics of Kenyan *Dodonaea viscosa* Jacq (Sapindaceae) Populations

**DOI:** 10.3390/molecules25184130

**Published:** 2020-09-10

**Authors:** Magrate M. Kaigongi, Catherine W. Lukhoba, Purity J. Ochieng‘, Malcolm Taylor, Abiy Yenesew, Nokwanda P. Makunga

**Affiliations:** 1Department of botany, School of Biological Sciences, College of Biological and Physical Sciences, University of Nairobi, Nairobi P.O. Box 30197-00100, Kenya; clukhoba@uonbi.ac.ke; 2Kenya Forestry Research Institute, Nairobi P.O. Box 20412-00200, Kenya; 3Department of Chemistry, School of Physical Sciences, University of Nairobi, Nairobi P.O. Box 30197-00100, Kenya; jaelpurityochieng@gmail.com (P.J.O.); ayenesew@uonbi.ac.ke (A.Y.); 4Central Analytical Facilities (CAF), Stellenbosch University, JC Smuts Building, Van Der Byl Rd, Stellenbosch Central, Stellenbosch 7600, South Africa; mtaylor@sun.ac.za; 5Department of Botany and Zoology, Stellenbosch University, Private Bag X1, Matieland 7602, South Africa

**Keywords:** *Dodonaea viscosa*, African traditional medicine, antimicrobial activity, chemosystematics, hopbush, metabolomics, natural products chemistry, phenolics, terpenoids

## Abstract

*Dodonaea viscosa* Jacq (Sapindaceae) is a medicinal plant with a worldwide distribution. The species has undergone enormous taxonomic changes which caused confusion amongst plant users. In Kenya, for example, two varieties are known to exist based on morphology, i.e., *D. viscosa* var. *viscosa* along the coast, and *D. viscosa* var. *angustifolia* in the Kenyan inland. These two taxa are recognized as distinct species in some reports. This prompted us to apply metabolomics to understand the relationship among naturally occurring populations of *D. viscosa* in Kenya, and to identify compounds that can assist in taxonomic delineation of the different varieties of *D. viscosa* from different parts of Kenya. The phytochemical variability of Kenyan *D. viscosa* var. *angustifolia* populations collected from four different geographical regions (Nanyuki, Machakos, Nairobi, and Narok) and one coastal *D. viscosa* var. *viscosa* (the Gazi) were analyzed by LC-MS using a metabolomics-driven approach. Four known compounds, two diterpenoids (dodonic acid (**1**), hautriwaic acid lactone (**3**), and two flavonoids (5,7,4′,5′-tetrahydroxy-3,6,2′-trimethoxyflavone (**2**) and catechin (**4**)) were isolated and purified from the Gazi coastal collection. The presence of these compounds and their relative abundance in other populations was determined by LC-MS analyses. Multivariate statistical analyses of LC-MS data was used for the visualization of the patterns of variation and identification of additional compounds. Eleven discriminant compounds responsible for separating chemometric clusters were tentatively identified. In an antimicrobial assay, hautriwaic acid lactone (**3**) and catechin (**4**) were the most active compounds followed by the extract from the coastal (Gazi) population. The clustering pattern of the five populations of *D. viscosa* suggested that the metabolite profiles were influenced by geo-environmental conditions and did not support the current classification of *D. viscosa* based on morphology. This study disputes the current classification of *D. viscosa* in Kenya and recommends revision using tools such as molecular phylogenetics.

## 1. Introduction

*Dodonaea viscosa* Jacq (Sapindaceae) is a globally distributed medicinal plant and its extracts have been shown to have anticancer activity against breast [1] and lung [2] cancer cell lines. The species has well established traditional uses that include treating fever, sore throat, burns, wounds, hemorrhoids, gout, indigestion, ulcers, diarrhea, constipation, trachoma, fractures, rheumatism, malaria, and snake bites [3]. In Ethiopia, the plant has been reported to treat a batch of diseases such as lymphatic swellings and cancer [4]. *D. viscosa* has also been reported to possess antimicrobial, anti-inflammatory, and antioxidants effects; amongst others [5,6,7]. Its therapeutic activity is likely associated with the synergistic effect of several constituents [8]. Apart from its medicinal uses, the species is used for construction purposes and as an ornamental plant across the globe [9]. The wide medicinal uses of *D. viscosa* has led to identification of several phytochemicals. The polyphenolic content and saponins form a substantial fraction of the phytochemical extracts of *D. viscosa*, making it a species with exceptional bioactivity [9,10,11].

Harrington and Gadek [12] indicated that *D. viscosa* was highly variable and studies have mainly used morphological characters to define different subspecies of the taxa. There are seven subspecies recognized in Australia, the place believed to be the center for its evolution, namely *D*. *viscosa* subsp. *angustifolia*, *D*. *viscosa* subsp. *angustissima*, *D*. *viscosa* subsp. *burmanniana*, *D*. *viscosa* subsp. *cuneata*, *D*. *viscosa* subsp. *mucronata*, *D*. *viscosa* subsp. *spatulate,* and *D. viscosa* subsp. *viscosa*). Each of these subspecies has a distinct habitat and can handle varying degrees of drought. Major differences that exist among these subspecies are related to growth form, morphological characteristics, and distribution [13].

In Kenya, the following two varieties are recognized: (1) *D. viscosa* var. *viscosa,* a thick bush/shrub that grows to 3–4 m with a fruit capsule that is usually white or brown with two large wings and bisexual flowers. The leaves are somewhat larger than *D. viscosa* var. *angustifolia*. This variant is referred to locally as Mkaa pwani (Swahili) and is more confined to the coastal parts of Kenya. (2) *D. viscosa* var. *angustifolia* (L.f.) Benth. is a shrub or small tree 1–6 m with pinkish or reddish, two to three winged fruits, and unisexual yellow-green flowers, with a much wider distribution, occurring inland in Kenya [14]. Because the latter is found in many different parts of Kenya, it has many different common names and it is known in local languages as Hidesa (Boran), Kithongoi/Muthongoi (Kamba), Murema muthua (Kikuyu), Muendu (Luhya), Oking’ (Luo), Olgeturai/Oltuyesi (Maasai), Tobolokwo (Pokot), Tombolokwa (Sabaot), Msidu (Taita), and Tabilikuet (Tugen) [14,15]. The wide distribution in Kenya has led to distinctive populations being classified as separate species [14], a practice that has further compounded the taxonomy of this taxon. In Kenya, scientific studies have mainly focused on the phytochemistry of *D. viscosa* var. *angustifolia* [16]. Despite numerous studies that have provided insights into the pharmacological potential of the plant, no chemical data or biomarkers exist on Kenyan populations to assist with delineation of its biogeographical distribution [17].

Metabolomics approaches have been applied in other species as chemotaxonomic tools because of their high resolution power in providing insights into biogeographic trends, in terms of plant diversification and evolution [18,19,20]. Furthermore, obtaining metabolomic fingerprints has wide applications in natural product research, with many studies reporting on the use of metabolomic analysis for the quality control of medicinal plants, the manufacturing of phytopharmaceutical products [21,22,23], and, as part of the drug discovery pipeline where novel bioactive compounds are characterized [24]. The aim of this study was to utilize metabolomics-based chemotaxonomy to solve the existing challenges in the taxonomy of *D. viscosa* plants found in Kenya (Figure 1 and Table 1) and to determine whether unique chemical markers could be identified. Thereafter, these could be utilized as distinguishing characters that could aid in the botanical classification of this plant. Additionally, isolation of compounds from one of the populations was undertaken with the purpose of generating chemical standards that could be employed as biomarkers to define qualitative and quantitative intra- and interpopulation differences. As a proof-of-concept, relative abundance analysis of the purified compounds in the other populations was done. In vitro antimicrobial analysis of these compounds and the extracts from the five populations was conducted using both Gram-positive and -negative bacteria and chosen fungi of economic importance to determine the best population in terms of tested antimicrobial activity, as well as to compare the activity of the populations with the isolated compounds.

## 2. Results and Discussion

### 2.1. Identification of Compounds

From the extracts, 313 chemicals were present in all the chromatograms assessed, but only 55 compounds were putatively identified and were comprised of phenolics, terpenoids, and saponins, several of which are described here for the first time in *D. viscosa* (Table 2). Some of the chemicals that were common among the samples include: isorhamnetin, aromadendrin, and chlorogenic acid. The relative abundances of the identified compounds in each population are illustrated in the form of a heat map (Figure 2) in which some compounds are found to be more concentrated (light red in color) in specific populations and less concentrated in other collections (light green in color) with regards to their relative abundances.

### 2.2. Chemotype Variation

Principal component (PC) 1 and 2 accounted for 28% and 17.2%, respectively, of the variance (Figure 3A). Principal component 1 usually accounts for the major variation on the x-axis while PC 2 accounts for minor variation in the y-axis. According to the two PC scores, the Nanyuki and Machakos clusters fell in the same quadrat and appeared to cluster closer to each other in addition to production of similar metabolites in almost equal relative amounts leading to our choice of using an orthogonal partial least square-discriminant analysis (OPLS-DA) test to further separate them. The OPLS-DA test is suitable for diagnosing the differences between two groups or systems which are closely related, as it has the potential to reduce the dimensionality of metabolomics datasets [30]. Eleven metabolites were identified as the discriminants of the five populations of *D. viscosa* (Table 3 and Figure 3B). The majority of these chemicals belong to the flavonoid class of compounds and dodonic acid, ent-16j-hydroxy-labdan-3a, 8b-dihydroxy, 13(14)-en-15, 16-olide, hypophylin E, and terpenticin are from the diterpenoid group.

From the score plot of the OPLS-DA test (Figure 4A) it was apparent that the Machakos and Nanyuki collections were distinct from each other based on chemical composition. The members within the Machakos population differentiated into two distinct sub-clusters, suggesting inherent chemical bias within this population of plants (Figure 4A), similar to the Narok population in Figure 3A. This sub-clustering could be due to intraspecific hybridization, with members from populations occurring in adjacent ecological zones. Intraspecific hybridization is important for developing environmentally adaptive traits [31]. The OPLS-DA analysis (Figure 4B) revealed that sets of unique chemicals which were absent in the Nanyuki group were present in the collections from the Machackos area and vice versa. The chemical discriminants identified in the Nanyuki population were santin, pinocembrin, and dodonic acid, whereas those from the Machakos region were kumatakenin, penduletin, and 5,7,4′-trihydroxy-3′-(4-hydroxy-3-methylbutyl)-5′-prenyl-3,6-dimethoxyflavone. Therefore, the OPLS-DA was powerful in diagnosing the differences between two systems or groups by determining the variables with high discriminatory power [32].

Absolute quantification of discriminants identified from the five different collections of *D. viscosa* plants (Table 3) showed that the Nanyuki population had the highest relative abundance of six chemical markers (santin, pinocembrin, dodonic acid, p-coumaric acid ethyl ester, terpenticin 2, and hypophyllin E) (Figure 5). The Gazi coastal and Nairobi plants each had higher levels of three and two chemicals, respectively, that discriminated the collections based on site. For the coastal types, two dimers of terpentecin and terpenticin 1 were differentially accumulating. As opposed to this, the Nairobi group had kumatakenin and ent-16j-hydroxy-labdan-3a,8b-dihydroxy,13(14)-en-15,16-olide as the chemicals that were upregulated.

Extensive natural disparity in phytochemical profiles occurs between and within species of plants as an adaptation measure to different abiotic and biotic environments [32]. The most important players in the biosynthesis and accumulation of secondary metabolites include genetics, epigenetics, morphogenetic, ontogenic, and environmental factors [33]. The PCA, as shown in Figure 3A, indicates that Nanyuki and Machakos populations of *D. viscosa* are closely related while the plants from the Nairobi and Narok areas are from a similar chemical lineage. The current morphological classification of the genus *Dodonaea* in Kenya shows that the coastal population is believed to be composed of *D. viscosa* subsp *viscosa,* whereas plants from the other locations are composed of *D. viscosa* subsp *angustifolia* [14]. The data of this study do not offer support for this idea. Instead, our data suggest that chemicals produced by the plants from different areas are related to the geographical and environmental conditions prevailing at the sites where each population was collected [34].

The production of specialized metabolites in plants is meant to help plants fight natural enemies, as well as counteract with other forms of natural biotic and abiotic stress including those created through anthropogenic activities such as pollution [35]. The environmental conditions prevailing in the Machakos and Nanyuki areas include drought together with high daytime temperatures and very low temperatures at night [36]. Hypothetically, it is possible that such differences in geoclimatic conditions can lead to variable chemistry that enables the plant to adapt to stress. Certainly, the Nairobi and Narok populations are faced with natural enemies such as browsers (Narok), as well as environmental stress such as pollution (Nairobi population) [35]; and thus, the group of chemicals produced by these populations is possibly meant to counter pollution and browsing effects experienced by these populations.

Both biotic and abiotic factors such as drought, browsing, salinity, heat, and air pollutants lead to heightened production of reactive oxygen species (ROS) in plants [37]. Studies have demonstrated the importance of non-enzymatic systems of intracellular antioxidant defense machinery in countering a variety of stresses. These non-enzymatic systems are mediated by antioxidants such as flavonoids [38]. Plants from the Gazi coast were found to have a different group of discriminants. Although this is speculative, it can be argued that these chemicals facilitate environmental adaptations specially to counter high temperatures and salinity [38,39]. The temperatures in the Gazi region can vary from 24 to 30 °C and because of a coastal locality, the soils are likely to be more saline. Flavonoids and diterpenoids such as catechin and hautriwaic acid lactone are implicated in stress responses in plants, and thus it is not surprising that these compounds were much higher in the plants from this hot and humid coastal region. This could possibly explain why samples from this area have close chemical similarities to the Nanyuki and Machakos cluster as compared with the plants growing in the Nairobi and Narok locations.

It can be argued that the metabolite production by the different populations of *D. viscosa* in Kenya is possibly based on prevailing environmental conditions [40]. This is probably influenced by epigenetic mechanisms which enhance phenotypic plasticity possibly as an adaptation to address the different levels of environmental stresses and in such cases, this can be transmitted successfully to the next offspring for numerous generations ultimately leading to transgenerational changes that evolve over time [32]. We propose this idea as the plants that are currently classified as *D. viscosa* subsp *viscosa*, growing in the Gazi coast, are in fact more similar in their phytochemical composition to the Nanyuki and Machakos populations that are composed of plants that have been designated to be *D. viscosa* subsp. *angustifolia*. On the contrary, the Narok and Nairobi populations, which are composed of *D. viscosa* subsp *angustifolia,* were found to have no relatedness with Nanyuki and Machakos populations, which were also composed of *D. viscosa* subsp *angustifolia*. This is the first report on the relatedness of *D. viscosa* populations in Kenya, and based on the data presented in this study, a revision of the taxonomic classification is needed. Other molecular based techniques, such as a phylogenetic analysis could provide new insights into the taxonomy of *D. viscosa* in Kenya.

### 2.3. Secondary Metabolites Isolated from Gazi Coastal Population of Dodonaea viscosa

The coastal population of *D. viscosa* CH₂Cl₂/CH_3_OH (1:1) extract yielded four known compounds (Appendix A) which were identified by MS and NMR (using COSY, HSQC, NOESY, and HMBC) analyses (see Appendix A) as dodonic acid (**1**) C_20_H_28_O_4_ [17], 5,7,4′,5′-tetrahydroxy-3,6,2′-trimethoxyflavone (**2**) C_18_H_16_O_9_) [41]; hautriwaic acid lactone (**3**) C_20_H_26_O_3_) [17]; and catechin (**4**) C_15_H_14_O_6_ [42]_._ This is the first report on the isolation of 5,7,4′,5′-tetrahydroxy-3,6,2′-trimethoxyflavone (**2**) from *D. viscosa*.

The relative abundance of dodonic acid (**1**) was high in all the five populations, whereas hautriwaic acid lactone (**3**) was present in Nanyuki, Machakos and Gazi coastal collections (Figure 6). In addition, the relative abundance of 5,7,4′,5′-tetrahydroxy-3,6,2′-trimethoxyflavone (**2**) was high in the Gazi coastal plants as compared with the other groups.

### 2.4. Antimicrobial Activity of the Isolated Compounds and the Extracts

Many plant secondary metabolites are known to possess a wide range of biological activities such as antifungal, antioxidant, antibacterial, antidiabetic, and anticancer activities [43,44]. *D. viscosa* has been reported to possess antimicrobial activity against microbes of economic importance [45,46,47,48]. Diterpenoids are known to possess antimicrobial and antioxidant potencies [49,50]. This prompted us to test the antimicrobial activity of the four isolated compounds from this plant against both Gram-positive (MRSA and *S. aureus*), Gram-negative (*E. coli*), and fungi (*C. albicans*) (Table 4). On the one hand, of the two diterpenoids isolated, dodonic acid (**1**) exhibited low activity against *S. aureus* and *E. coli* and was inactive against MRSA and *C. albicans*. On the other hand, hautriwaic acid lactone (**3**) with a similar structure to dodonic acid (**1**), except for the presence of a lactone ring, showed good activity against all the tested microbes. The MIC values for both *S. aureus* and *E. coli* were 1.95 µg/mL, MRSA 62.5 µg/mL, and 7.81 µg/mL for *C. albicans*. These results indicate the importance of a lactone moiety in the inhibition of microbial growth [51]. The report of antimicrobial activity of hautriwaic acid lactone is in agreement with the work of Omosa et al. [17] who found that this compound had good antifungal activity against *S. cerevisiae* with a MIC value of 7.8 μg/well, probably due to its lipophilic nature.

Plant flavonoids have been reported to exhibit a wide range of biological activities including antibacterial, antioxidant, anticancer, and antifungal [52,53]. Interestingly, catechin (**4**), a flavonoid purified from *D. viscosa,* inhibited the growth of the four tested microbes with MIC values for *S. aureus* and *E. coli* of 3.91 µg/mL, and 7.81 µg/mL for MRSA and *C. albicans.* The antimicrobial activity of catechin is inconsistent with those reported in other studies [54,55,56,57]. 5,7,4′,5′-tetrahydroxy-3,6,2′-trimethoxyflavone (**2**), a flavonoid with similar oxygenation pattern with catechin except for the absence of OH groups at carbon 3 and 3′ did not inhibit the growth of any of the microbes tested, confirming the importance of hydroxyl groups in those carbons for antimicrobial activity in flavonoids. Hydroxyls at special sites on the aromatic rings of flavonoids are known to improve their biological activities, whereas methylation of the active OH groups generally decreases such activities [58]. The coastal Gazi population extracts were the most active followed by the extracts from the Nanyuki plants, whereas the Narok plant collections had the least active extracts. The activity of the coastal plants against *S. aureus* and *E. coli* was similar to catechin (**4**) while the Nanyuki extract and hautriwaic acid lactone (**3**) were similar to each other in the inhibition of the growth of *C. albicans*.

## 3. Materials and Methods

### 3.1. Collection of Plant Materials

*D. viscosa* plants samples were collected from different natural origins across five localities in Kenya. Four collection sites, i.e., Nanyuki, Machakos, Karura, and Narok were from Kenyan inland regions and were composed of *D. viscosa* var *angustifolia*. The Gazi collection site is along the Kenyan coast and the plants collected there were composed of *D. viscosa* var. *viscosa*. Three samples from different individual plants were taken from each population. Sampling of specimens was conducted in April 2018. Voucher specimens were deposited at the Nairobi University Herbarium after the confirmation of the taxonomic identities by Dr C. Lukhoba (a botanist at the University of Nairobi, Nairobi, Kenya). Details of voucher specimens and geographic collection localities are as shown in Table 1. The plants’ materials were washed, cut into smaller leaf pieces (size 2 cm), and dried away from direct sunlight, at room temperature for 15 days. This was followed by grinding each sample separately by hand, in a pestle and mortar.

### 3.2. Extraction for Metabolite Profiling

A similar protocol to that of Albrecht et al. [23] was used in this study. Briefly, 50 mg from each sample was extracted with 50% (*v*/*v*) acetonitrile (Romil far UV grade, Microsep, Johannesburg, South Africa) containing 0.1% (*v*/*v*) formic acid (Sigma-Aldrich, Johannesburg, South Africa). In all cases, ultrapure water (MilliQ) was used as a diluent. This study was carried out at the Central Analytical Facilities (CAF), Stellenbosch University, Stellenbosch, South Africa.

#### Standards

Two standards were obtained from Sigma-Aldrich (Johannesburg, South Africa), i.e., quercetin and catechin, and were analytically weighed and dissolved in dimethyl sulfoxide (DMSO) and diluted in methanol to set up a calibration series of 10, 20, 50, 100, and 200 ppm.

### 3.3. Liquid Chromatography Mass Spectrometry (LC-MS) Analysis

Analysis was performed using a Waters Synapt G2 quadrupole time-of-flight High Definition Mass Spectrometer (Milford, MA, USA), linked to a Waters Acquity ultra-performance liquid chromatograph (UPLC) and Acquity photo diode array (PDA) detector. Ionization was achieved with an electrospray source using a cone voltage of 15 V and capillary voltage of 2.5 kV in negative mode. Nitrogen was used as the desolvation gas at 650 l h^−1^ and the desolvation temperature was set to 275 °C. A Waters UPLC BEH C18 column (2.1 × 100 mm, 1.7 μm particle size) was used and 3 μL was injected for each analysis. The gradient started with 100% using 0.1% (*v*/*v*) formic acid (solvent A) and this was kept at 100% for 0.5 min, followed by a linear gradient to 22% acetonitrile (solvent B) over 2.5 min, 44% solvent B over 4 min, and finally to 100% solvent B over 5 min. The column was subjected to 100% solvent B for an additional 2 min. Then, the column was re-equilibrated over 1 min to yield a total run time of 15 min. A flow rate of 0.4 mL min^−1^ was applied. The MSE mode was used to acquire the data with a low collision energy scan followed by a high collision energy scan with collision energy ramped from 20 to 60 to obtain both molecular ion [M − H]^−^ and fragment data at the same time. For each biological replicate of the materials collected, there were three technical replicates analyzed.

### 3.4. Extraction and Isolation of Compounds: Gazi Coastal Population

Ground material of (800 g) of *Dodonaea viscosa* Gazi coastal plants was exhaustively and sequentially extracted with a sufficient amount of methanol (CH_3_OH)/dichloromethane (CH_2_Cl_2_) in the ratio of 1:1, for 72 h. The resultant extract was filtered to remove debris and the solvent was removed in vacuo using a rotary evaporator and resulted in 220 g of the crude extract. A portion of this extract (50 g) was adsorbed in 100 g of silica gel 60 (70–230 mesh) and subjected to column chromatography on silica gel 60 (70–230 mesh) (700 g) packed under 100% *n*-hexane. The column was eluted with 100% *n*-hexane followed by increasing polarity of EtOAc in *n*-hexane starting from 1% to 100% of EtOAc, followed by CH_3_OH in EtOAc in increasing percentages. A total of 240 fractions, each 400 mL, were collected and combined on the basis of TLC (on silica gel 60 F254, Merck) analysis. Elutions made at 1–15% of EtOAc in *n*-hexane did not yield any compound but resulted in oily substances. Fractions eluted between 15% and 30% of EtOAc in *n*-hexane were combined and washed with *n*-hexane/EtOAc (3:2) to give a white amorphous solid **1** (100 mg). Elutions obtained at 40% of EtOAc in *n*-hexane were combined and washed with *n*-hexane/EtOAc (3:2) to yield a yellow powder of **2** (11 mg). Fractions obtained between 40% and 100% were further combined to yield 10 g, and further separated by column chromatography over silica gel using *n*-hexane containing increasing polarity of EtOAc and yielded a white amorphous compound **2** (29 mg) at 30% of EtOAc in *n*-hexane elution.

### 3.5. Characterization of Pure Compounds

This was carried out at the University of Potsdam (Brandenburg, Germany) and the Central Analytical Facility, Stellenbosch University (Stellenbosch, South Africa). Structures of the purified compounds were determined through ^1^H-, ^13^C-NMR using Bruker Avance 500 MHz spectrometers, and HMBC, HSQC, COSY, and NOESY spectroscopy using standard Bruker software. Chemical shifts were measured in ppm in δ values relative to the internal standard tetramethyl silane (TMS).

### 3.6. Antimicrobial Activity of the Isolated Compounds

All the pure compounds and extracts from the five populations were tested for antimicrobial activity against the following: *Staphylococcus aureus* (ATCC 29213), *Escherichia coli* (ATCC 25922), *Candida albicans* (ATCC 10231), and un-typed isolate of methicillin resistant *Staphylococcus aureus* (MRSA). The microbes were obtained from the Centre for Microbiology Research (CMR), Kenya Medical Research Institute (KEMRI). The stock solutions of the compounds were prepared by separately dissolving 1 mg of each compound in 1 mL of 10% DMSO to make 1000 µg/mL. The minimal inhibition concentration (MIC) of the chemical isolates from the Gazi coast population of *D. viscosa* was determined using a broth dilution technique. All the compounds and extracts were diluted serially, ranging from 1000 to 0.011 µg/mL in 96 micro well plates. For each dilution, 1 mL of 24-hour and 72-hour cultured bacteria and fungi, respectively, were added, adjusted to McFarland turbidity [59,60], and placed in an incubator at 37 °C for 24 h for bacteria, and 30 °C for 72 h for fungi. The experiment was carried out in triplicate. The lowest dilution for each compound with invisible microbial growth was recorded as MIC. This was confirmed by absence of turbidity after inoculating into agar followed by incubation for 24 and 72 h for bacteria and fungi, respectively [61]. Omacilin and fluconazole obtained from Sigma-Aldrich (St. Louis, Missouri, USA) served as positive controls for bacteria and fungi, respectively, whereas 10% DMSO was used as a negative control.

### 3.7. Statistical Analysis

Three biological replicates and three technical replicates were injected randomly to the LC-MS machine for each test sample. Untargeted analysis of the data was performed using MarkerLynx (part of MassLynx 4.1, Waters, Milford, USA) to generate a matrix of accurate mass and retention time (AMRT) features uploaded to MetaboAnalyst, a free online software platform, to perform the chemometric analysis. Targetlynx, another application within Masslynx, was used to quantify the various tentatively identified metabolites in a relative fashion, against calibration curves established using either catechin or quercetin calibration standards. The principal component analysis (PCA) model with pareto scaling was used as an unsupervised multivariate cluster technique. Pareto scaling increases the contributory effects of low concentration metabolites but does not amplify noise and artefacts linked to metabolomics data [62]. Therefore, this simplifies the interpretation of loading score plots [63]. Interrelationships were revealed through PCA based on three components (MS data, relative abundances, and spatial distribution) for the test materials. The instrument variation was negligible and replicates clustered close to each other. Where members from different sites appeared to cluster together, orthogonal partial least squares discriminant analysis (OPLS-DA) was performed to further separate them. The accurate mass obtained for the molecular ions [M − H]^−^ was used to calculate the possible molecular formula, considering a maximum deviation between observed and calculated mass of 5 ppm. The molecular formula, accurate mass, and MSE fragmentation spectra, were used as entry data for MetFrag β-database analysis [64] to identify the compounds where possible.

## 4. Conclusions

This study does not support the current morphological classification of the genus *D. viscosa* in Kenya, where the coastal population is said to be composed of *D. viscosa* subsp *viscosa*, whereas the in-country are classified as *D. viscosa* subsp *angustifolia.* The metabolomics data indicate the Gazi coast (*D. viscosa* subsp *viscosa*) population to be chemically similar to the Nanyuki and Machakos populations that are composed of plants that have been designated to be *D. viscosa* subsp *angustifolia*. Therefore, a revision of the taxonomic classification is recommended as the Narok and Nairobi collections had no relatedness with Nanyuki and Machakos plants that are all currently regarded as subspecies *angustifolia*. Further characterization of these populations is needed and a molecular phylogenetic analysis would provide new insights into the taxonomy of *D. viscosa* in Kenya. From the four compounds isolated (two diterpenoids and two flavonoids), hautriwaic acid lactone (**3**), and catechin (**4**) were the most active in terms of antimicrobial activity. This is the first report on the isolation of 5,7,4′,5′-tetrahydroxy-3,6,2′-trimethoxyflavone (**2**) from this species. We also recommend studies of different pharmacological activities (both in vitro and in vivo) to determine whether the *D. viscosa* populations are different in terms of their bioactivity.

## Figures and Tables

**Figure 1 molecules-25-04130-f001:**
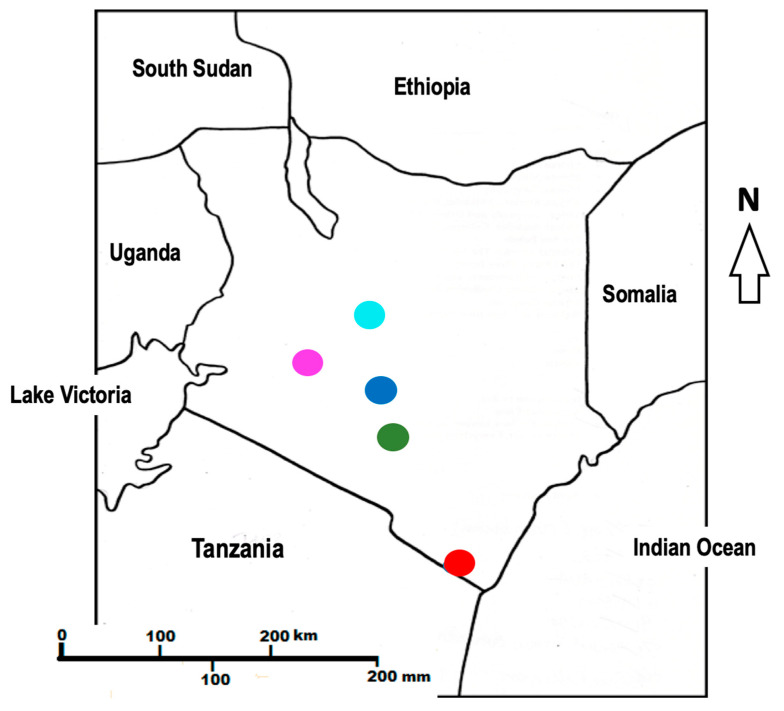
Map of Kenya showing the following sampling points: Gazi coast (red), Machakos (green), Nairobi (blue), Nanyuki (cyan), and Narok (purple).

**Figure 2 molecules-25-04130-f002:**
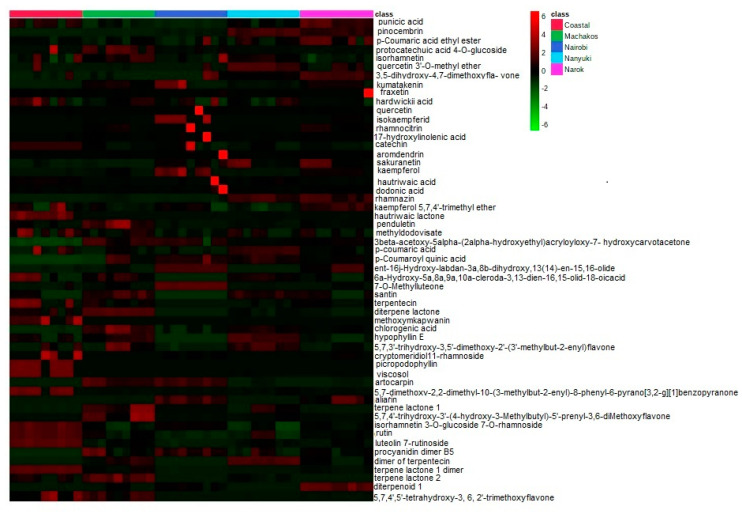
Heat map showing the relative abundance of the identified compounds in each population.

**Figure 3 molecules-25-04130-f003:**
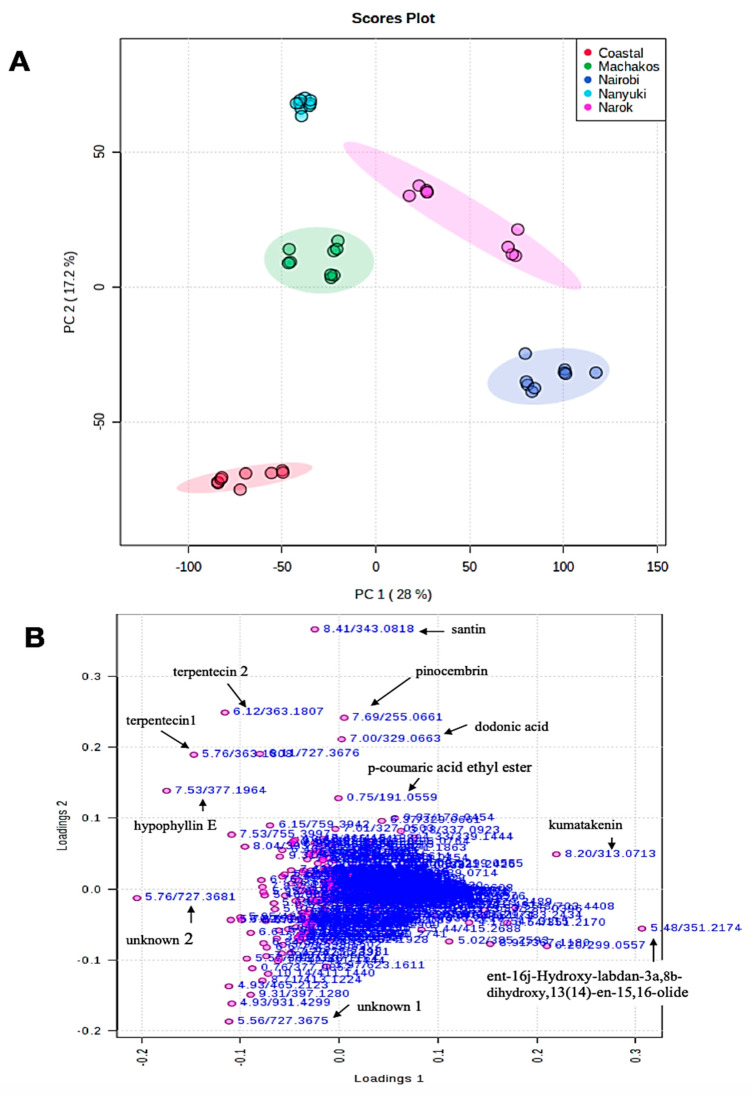
(**A**) Principal component analysis (PCA) score plot showing variation of *D. viscosa* plants from different natural population five different locations in Kenya, i.e., coastal region (red color), Machakos (green color), Nairobi (blue color), Nanyuki (cyan color), and Narok (purple color); (**B**) PCA loadings plot of compounds that influenced the differentiation of collections into separate clusters.

**Figure 4 molecules-25-04130-f004:**
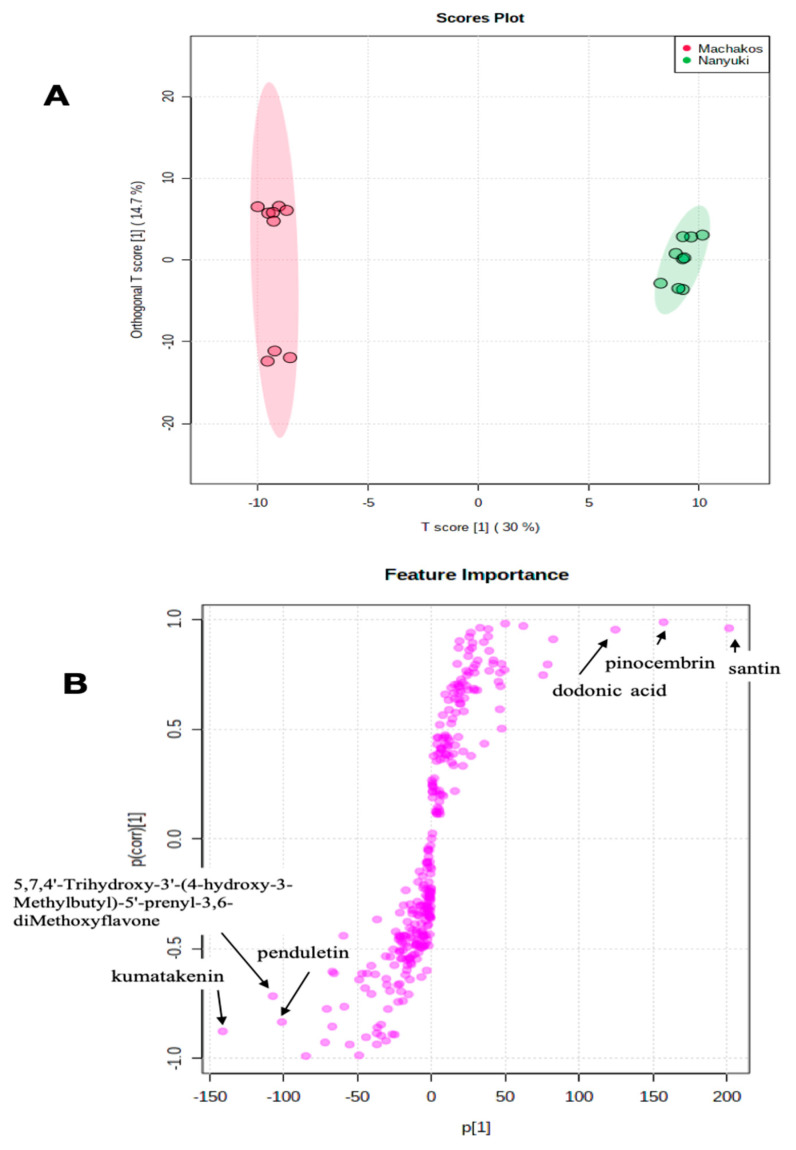
(**A**) Orthogonal partial least squares discriminant analysis (OPLS-DA) for *Dodonaea viscosa* Nanyuki and Machakos populations. The OPLS-DA test separation of Machakos and Nanyuki populations of *D. viscosa* was based on T score 1 which was 30% separating Machakos population on the left and the Nanyuki population on the right; (**B**) Loadings of OPLS-DA of LC-MS spectra for *D. viscosa* Nanyuki and Machakos populations showing the compounds that are important discriminants.

**Figure 5 molecules-25-04130-f005:**
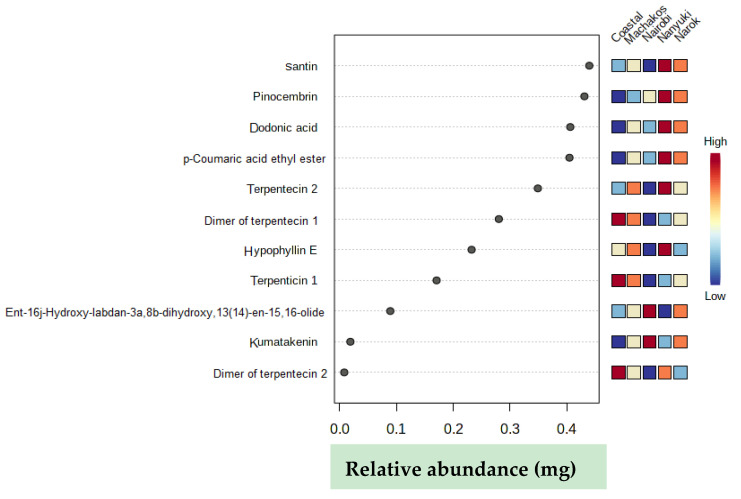
Relative abundance of chemical discriminants in different *D. viscosa* populations. The dark red color shows high relative abundance of the individual chemical in each population while the dark blue color implies a low relative abundance.

**Figure 6 molecules-25-04130-f006:**
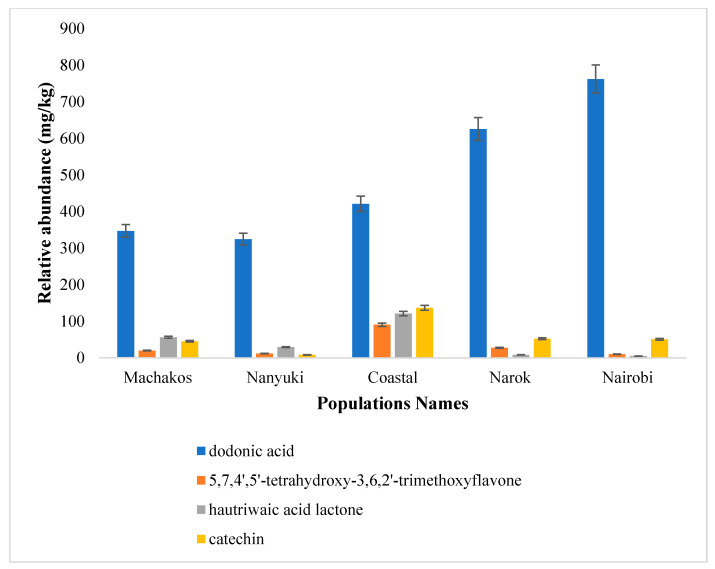
Concentrations (mg/kg) of the isolated compounds (dodonic acid (**1**), 5,7,4′,5′-tetrahydroxy-3,6,2′-trimethoxyflavone (**2**), hautriwaic acid lactone (**3**), and catechin (**4**) in the five Kenyan *D. viscosa* populations.

**Table 1 molecules-25-04130-t001:** Geographical collections of *D. viscosa* showing details of specific collections, as well as voucher numbers.

No.	Locality Name	Specific Location	GPS (S)	GPS (E)	Voucher No
1.	Machakos	Kaani ka itheu	1° 29′ 55.8″	37° 21′ 58.2″	MK2/2018
2.	Nanyuki	Kahurura	0° 02′ 06.0″	37° 07′ 49.4″	MK4/2018
3.	Coast	Gazi	4° 25′ 29.1″	39° 30′ 22.5″	MK5/2018
4.	Nairobi	Karura Forest	1° 14′ 43.6″	36° 50′ 17.4″	MK8/2018
5.	Narok	Maasai Mara Reserve	1° 28′ 36.5″	35° 05′ 33.3″	MK9/2018

**Table 2 molecules-25-04130-t002:** List of compounds tentatively identified in *D. viscosa* showing retention times, detected [M − H]^−^ ion, elemental composition, MSE fragments, PPM error, and UV absorbance.

	Experimental m/z [M − H]^−^	Retention Time (min)	Formula	PPM Error	MSE Fragments	UV (nm)	Proposed Compound	References
1.	341.1081	0.75	C_21_H_25_O_4_	1.2	341.1078, 173.0419, 377.0883,515.1646	weak	Methyl dodovisate B	[25]
2.	191.0559	0.75	C_10_H_7_O_4_		191.0557, 377.0834, 379.0815, 719.1973	weak	p-Coumaric acid ethyl ester	[26]
3.	315.0507	1.32	C_16_H_12_O_7_	0	315.0504, 300.0272, 151.0038, 107.0134	weak	Isorhamnetin	[8,17]
4.	315.0711	2.41	C_13_H_15_O_9_	−1.9	315.0612, 153.0178, 109.0285	weak	Protocatechuic acid 4-*O*-glucoside	First report *
5.	315.0718	2.41	C_20_H_27_O_3_	0.8	315.0724, 316.0713, 327.0630, 463.0933	weak	Hardwickiic acid	[25]
6.	207.0292	2.66	C_10_H_7_O_5_	−1.4	207.0287, 163.0394, 119.049, 165.0393	282	Fraxetin	First report *
7.	353.0865	2.98	C_16_H_17_O_9_	−2.3	353.0872, 191.0555, 173.0447, 119.0487	322	Chlorogenic acid	First report *
8.	577.1351	3.04	C_30_H_25_O_12_	0.9	577.1346, 191.0556, 125.0232, 289.071	280	Procyanidin dimer B5	First report *
9.	289.0719	3.21	C_15_H_13_O_6_	1.4	289.0715, 105.0195, 161.0249, 267.0509	279	Catechin	[8]
10.	337.0923	3.28	C_16_H_17_O_8_	−1.2	337.0915,173.045, 191.0553, 289.0701	281	p-Coumaric acid	[9]
11.	313.0234	3.42	C_20_H_25_O_3_	1.9	313.2361, 309.2002, 311.2331, 311.2128	279	Hautriwaic acid lactone	[25]
12.	431.1901	3.43	C_27_H_27_O_5_	−6.9	382.119, 300.0346, 153.0962	weak	5,7-Dimethoxy-2,2-dimethyl-10-(3-methylbut-2-enyl)-8-phenyl-6-pyrano[3 ,2-g][1]benzopyranone	First report *
13.	609.1458	3.7	C_27_H_29_O_16_	0.3	609.1367, 301.0345, 300.0277, 477.0683	weak	Rutin	[27]
14.	593.1502	3.9	C_27_H_29_O_15_	−1.0	593.1522, 285.0408	351	Luteolin 7-rutinoside	First report *
15.	623.1612	4.02	C_28_H_31_O_16_	1	315.0509, 300.028, 271.0279, 243.0317	weak	Isorhamnetin 3-*O*-glucoside 7-*O*-rhamnoside	First report *
16.	287.0551	4.32	C_15_H_12_O_6_	−1.9	287.2227, 329.2318, 327.2163, 288.2239	361	Aromadendrin	[11]
17.	339.1449	4.36	C_17_H_23_O_7_	2.13	287.0554, 176.0869, 121.0287	weak	3-β-acetoxy-5-α-(2-α-hydroxyethyl)acryloyloxy-7-hydroxycarvotacetone	First report *
18.	465.1913	4.93	C_24_H_33_O_9_	1.09	379.1883, 285.1477, 241.1612, 119.0352	340	Terpene lactone 1	First report *
19.	931	4.95	C_48_H_67_O_18_	−1.6	931,4309, 465.212, 241.1595, 285.1487	weak	Terpene lactone 1 dimer	First report *
20.	385.267	5.04	C_21_H_37_O_6_	−0.5	385.2659, 325.2376, 285.1486, 177.0921	weak	Cryptomeridiol-11-rhamnoside	First report *
21.	361.1647	5.46	C_20_H_25_O_6_	0.72	351.2182,3 17.17, 307.23, 243.18, 126.03	250	Diterpene lactone	First report *
22.	351.2171	5.48	C_20_H_31_O_5_	0.6	351.2170, 307.2278, 315.0502, 249.1857	weak	ent-16j-Hydroxy-labdan-3a,8b-dihydroxy,13(14)-en-15,16-olide	[28]
23.	363.181	5.76	C_20_H_27_O_6_	0.6	363.1711, 319.1913, 275.201, 259.1695	weak	Terpentecin 1	First report *
24.	727.3684	5.76	C_40_H_55_O_12_	−1.4	727.3680,363.180, 364.1838,275.2008	weak	Dimer of terpentecin	First report *
25.	285.0399	5.87	C_15_H_9_O_6_	0	285.0397, 241.1589, 151.0029, 242.162	266, 366	Kaempferol	[16]
26.	301.0711	5.93	C_15_H_9_O_7_	3.0	301.0700, 609.1830, 610.1893, 302.0850	251	Quercetin	[8]
27.	315.0021	5.97	C_30_ H_25_O_12_	1.1	315.2560, 293.2096, 316.2598, 249.1506	weak	Quercetin 3′-*O*-methyl ether	[9]
28.	363.1807	6.12	C_20_H_27_O_6_	0.6	363.1805, 319.1913, 275.201, 259.1695	weak	Terpentecin 2	First report *
29.	429.2489	6.13	C_23_H_26_O_8_	2.7	429.2496, 351.2181, 299.0559, 285.0429	261	Aliarin	[26]
30.	299.0557	6.2	C_16_H_11_O_6_	1.6	299.0557, 375.1808, 347.1865, 300.0591	weak	Isokaempferide	[17,27,29]
31.	327.2164	6.3	C_18_H_16_O_6_	1.8	327.2166, 328.2183, 325.1966, 313.1455	weak	Kaempferol 5,7,4′-trimethyl ether	[28]
32.	359.0764	6.58	C_21_H_27_O_5_	0.8	359.0974, 266.0666, 197.0446, 435.1606	weak	Methoxymkapwanin	[27]
33.	375.0723	6.88	C_18_H_16_O_9_	1.6	375.0722, 361.1641, 351.2161, 551.2051	280	5,7,4′,5′-Tetrahydroxy-3,6,2′–trimethoxyflavone	First report *
34.	329.0663	7.00	C_20_H_25_O_4_	0.2	329.0654, 361.1648, 347.1866, 418.2239	weak	Dodonic acid	[17]
35.	329.0661	7.07	C_17_H_13_O_7_	1.5	329.0660, 271.0249, 314.0432, 275.2003	277,337	Rhamnazin	[17]
36.	375.1808	7.13	C_21_H_28_O_6_	−4.46	345.17, 319.2, 259.17, 116.93	280	Terpene lactone 1	First report *
37.	343.0822	7.43	C_18_H_15_O_7_	−1.3	343.0817, 344.0853, 393.1919, 345.1689	279	Penduletin	[28]
38.	377.1957	7.56	C_21_H_29_O_6_	−1.9	377.1957, 345.1701, 301.1799, 189.1278	350	Hypophyllin E	First report *
39.	285.0764	7.62	C_16_H_12_O_5_	1.5	285.0403, 286.0438, 533.2025, 571.0843	261	Sakuranetin	[27]
40.	255.0662	7.78	C_15_H_11_O_4_	1.2	255.0660, 151.0035, 213.0554, 107.0135	288	Pinocembrin	[17]
41.	347.1857	7.91	C_20_H_27_O_5_	−0.3	347.1850, 303.1965, 285.1488, 241.1591	weak	(A)-6a-Hydroxy-5a,8a,9a,10a-cleroda-3,13-dien-16,15-olid-18-oicacid	[10]
42.	299.0560	7.96	C_16_H_11_O_6_	1.3	299.055, 271.0605, 65.0193, 284.0329	266,365	Rhamnocitrin	[17]
43.	313.0713	8.20	C_17_H_13_O_6_	0.7	313.0712, 314.0749 377.1953, 298.0486	361	Kumatakenin	[17]
44.	343.0812	8.41	C_18_H_15_O_7_	−1.7	343.0812, 313.0346, 301.1798, 270.0161	270,340	Santin	[17]
45.	413.1236	8.76	C_22_H_21_O_8_	0.5	413.1231, 368.0905, 331.1908, 161.0144	351	Picropodophyllin	First report *
46.	367.1180	8.91	C_21_H_19_O_6_	−0.5	352.0954, 323.0915, 297.0328, 269.044	267	7-*O*-methylluteone	First report *
47.	313.0720	8.98	C_17_H_13_O_6_	2.6	313.0720, 283.0252, 255.0296, 161.0255	267,347	3,5-Dihydroxy-4′,7-dimethoxyfla-vone	[17,27]
48.	331.1910	9.19	C_20_H_27_O_4_	0.3	331.1914, 397.1290, 332.1949, 398.1334	340	Hautriwaic acid	[17]
49.	397.1280	9.31	C_22_H_21_O_7_	−0.4	397.1291, 1169.585, 331.1901, 1156.5643	weak	5,7,3′-Trihydroxy-3,5′-dimethoxy-2′-(3′-methylbut-2-enyl)flavone	[26]
50.	483.2021	9.35	C_27_H_31_O_8_	0.2	483.2018, 453.1683,331.1906, 397.1288	341	5,7,4′-Trihydroxy-3′-(4-hydroxy-3-Methylbutyl)-5′-prenyl-3,6-diMethoxyflavone	[26]
51.	293.2117	9.58	C_18_H_29_O_3_	−1.4	293.2111,161.0242, 152.9944, 265.1489	267,347	17-Hydroxylinolenic acid	First report *
52.	321.2443	10.08	C_20_H_33_O_3_	−2.43	277.2537, 116.938	weak	diterpenoid	First report *
53.	411.1444	10.19	C_23_H_23_O_7_	0	411.1439, 396.1214, 331.1898, 265.1464	340	Viscosol	[28]
54.	435.1808	10.36	C_26_H_27_O_6_	−1.6	435.1806, 255.2322, 365.102, 161.0247	weak	Artocarpin	First report *
55.	277.2168	11.33	C_18_H_29_O_2_	0	277.2163, 116.9271, 265.1465, 152.9955	weak	Punicic acid	First report *

* First report means they have not been previously reported in *D. viscosa.* Literature references for compounds previously reported in *D. viscosa* are also indicated.

**Table 3 molecules-25-04130-t003:** List of discriminants identified in *D. viscosa* populations in this study showing retention time, detected [M − H]^−^ ion, elemental composition, as well as the population with highest relative abundance of each discriminant.

	Retention Time (min)	ESI negative [M − H]^− (*m/z*)^	Elemental Composition	Name	Population with Highest Concentration
1.	0.75	191.0559	C_10_H_7_O_4_	p-coumaric acid ethyl ester	Nanyuki
2.	5.48	351.2174	C_20_H_31_O_5_	ent-16j-hydroxy-labdan-3a,8b-dihydroxy,13(14)-en-15,16-olide	Nairobi
3.	5.76	363.1808	C_20_H_27_O_6_	terpentecin 1	coastal (Gazi)
4.	5.76	727.3681	C_40_H_55_O_12_	dimer of terpentecin 1	coastal (Gazi)
5.	6.11	727.3676	C_40_H_55_O_12_	dimer of terpentecin 2	coastal (Gazi)
6.	6.12	363.1807	C_20_H_27_O_6_	terpentecin 2	Nanyuki
7.	7.00	329.0663	C_20_H_25_O_4_	dodonic acid	Nanyuki
8.	7.53	377.1964	C_21_H_29_O_6_	hypophyllin E	Nanyuki
9.	7.69	255.0661	C_15_H_11_O_4_	pinocembrin	Nanyuki
10.	8.20	313.07	C_17_H_13_O_6_	kumatakenin	Nairobi
11.	8.41	343.0818	C_18_H_15_O_7_	santin	Nanyuki

**Table 4 molecules-25-04130-t004:** MIC values (µg/mL) for the four isolated compounds from the coastal population of *D. viscosa* and the extracts from the five populations against methicillin resistant *Staphylococcus aureus* (MRSA), *Staphylococcus aureus*, *Escherichia coli,* and *Candida albicans*.

Sample Name	Microbial Names			
	MRSA	*S. aureus*	*E. coli*	*C. albicans*
Dodonic acid (**1**)	>1000	500	500	>1000
5,7,4′,5′-Tetrahydroxy-3,6,2′ trimethoxyflavone (**2**)	>1000	>1000	>1000	>1000
Hautriwaic acid lactone (**3**)	62.50	1.95	1.95	7.81
Catechin (**4**)	7.81	3.91	7.81	3.91
Coastal (Gazi)	62.50	3.91	7.81	15.62
Machakos	125	7.81	15.62	31.20
Nairobi	>1000	31.25	62.50	>1000
Nanyuki	62.50	7.81	15.62	7.81
Narok	125	15.62	31.25	62.5
Omacilin	0.98	0.49	0.98	-
Fluconazole				1.95

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
