# Peer review of "LC-MS-Based Metabolomics for the Chemosystematics of Kenyan Dodonaea viscosa Jacq (Sapindaceae) Populations"

_molecules, 2020, doi:10.3390/molecules25184130_

Round 1

Reviewer 1 Report

The taxonomic classification of the medicinal plant Dodonaea viscosa Jacq (Sapindaceae) has been a complex task which causes serious confusion amongst all plant researchers. In Kenya two varieties of this medicinal plant namely var D. viscosa var viscosa along the coast, and D. viscosa angustifolia in the Kenyan inland are recognised as distinct species.

For this reason, the authors attempted to investigate by LC-MS using a metabolomics-driven approach the phytochemical variability of Kenyan D. viscosa populations collected from five different geographical regions (Nanyuki, Machakos, Nairobi, Narok) and one coastal (the Gazi)

Consequently, the authors isolated a series of compounds from one of the populations, in order to generate chemical standards that could be employed as biomarkers to define qualitative and quantitative intra- and inter-population differences.

Moreover, they isolated, purified and four known compounds, two diterpenoids (hautriwaic acid (1), hautriwaic acid lactone (3) and two flavonoids [5,7,4',5'-tetrahydroxy-3,6,2'-trimethoxyflavone (2) and catechin (4)] were isolated and purified from the Gazi, coastal collection. Thus, the relative abundance analysis of the purified compounds  was determined in the other populations and this was followed by in vitro antimicrobial analysis of these compounds was conducted using both Gram positive and negative bacteria and fungi of  economic importance.

The authors used multivariate statistical analyses of the LC-MS data which permitted the visualization of any patterns variation and the identification of additional compounds. Fifteen discriminant compounds responsible for separating chemo metric clusters were tentatively identified. Also, the clustering pattern of the five chosen populations of D. viscose indicated that the metabolite profiles, were only influenced by the geo- environmental conditions and did not support the current classification of D. viscosa based on morphology.

To sum up the obtained results, the authors have shown that the metabolomics data indicated the Gaze coast population to be chemically similar to the Nanyuki and Machakos populations that are composed of plants that have been designated to be D. viscosa subsp angustifolia

In conclusion, this study contradicts the current classification of D. viscosa in Kenya and recommends the need for a revision using tools such as molecular phylogenetics.

In general, this manuscript is well written.  I find that the analytical work used in this metabolomics study to be acceptable.

However, I would advise the authors to indicate better the original classification attributed to the studied D. viscosa populations collected in the five different geographical regions chosen, namely; the Nanyuki, Machakos, Nairobi, Narok) and one coastal (the Gazi). That is which one belong to var D. viscosa var viscosa and D. viscosa angustifolia

1.In page 5 of your pdf, lines 121-123, you have written;

Based on the two PC scores, the Nanyuki and Machakos cluster fell on the same quadrat and appeared to cluster closer to each other, leading to our choice of using orthogonal partial least square-discriminant analysis (OPLS-DA) test to further separate them

Query: When you indicate for example  the Nanyuki and Machakos cluster, you most probably know which  species these were?  This referee and most probably the reader of the journal will not know and will be left wandering.

  1. I presume that your fifteen discriminant compounds responsible for separating chemometric clusters which were tentatively identified are the one shown in .Table 2. List of compounds tentatively identified in D. viscosa showing retention time, detected [M-H]- ion, elemental composition literature references for compounds previously reported in D. viscosa,.

Query: You did not explain correctly how did you identified this series of “First Report” compounds.

Surely, you did not use only the molecular mass of the deprotonated molecule [M-H] - and retention time! This is not acceptable.

First you have to shown in this Table 2, the differences in ppm between your observed deprotonated molecules with that of the calculated value.

The identification of your deprotonated molecule should consider the presence of one heteroatom (Oxygen), mass error (within 0.1 m/z unit), and reasonable MS/MS fragmentation patterns which confirms the identity of the studied molecule. In addition, the double bond equivalent (DBE), should be calculated for the precursor ions to determine the number of unsaturation. Finally, you should determine that the values of DBE and C/O ratio for the identified formulas are consistent with the values reported before in literature.

Author Response

RESPONSE TO REVIEWER 1 COMMENTS

General comment: I would advise the authors to indicate better the original classification attributed to the studied D. viscosa populations collected in the five different geographical regions chosen, namely; the Nanyuki, Machakos, Nairobi, Narok) and one coastal (the Gazi). That is which one belong to var D. viscosa var viscosa and D. viscosa angustifolia

Response: All the Dodonaea viscosa populations collected from Nanyuki, Machakos, Narok and Nairobi represent the D. viscosa var angustifolia while the coastal population collected from Gazi resprents D. viscosa var viscosa

Point 1: In page 5 of your pdf, lines 121-123, you have written;

Based on the two PC scores, the Nanyuki and Machakos cluster fell on the same quadrat and appeared to cluster closer to each other, leading to our choice of using orthogonal partial least square-discriminant analysis (OPLS-DA) test to further separate them

Query: When you indicate for example the Nanyuki and Machakos cluster, you most probably know which  species these were?  This referee and most probably the reader of the journal will not know and will be left wandering.

Response 1: The two populations, Nanyuki and Machakos were composed of D. viscosa subsp angustifolia

Point 2: I presume that your fifteen discriminant compounds responsible for separating chemometric clusters which were tentatively identified are the one shown in .Table 2. List of compounds tentatively identified in D. viscosa showing retention time, detected [M-H]- ion, elemental composition literature references for compounds previously reported in D. viscosa,.

Query: You did not explain correctly how did you identified this series of “First Report” compounds.

Response 2: The information in Table 2 has been enriched with details of MSE fragments, PPM error and UV for each tentatively identified compound

Reviewer 2 Report

Overall, the manuscript proposed the interesting topic toward the specialist of the field and potential readers of the journal. Experimental design, analytical methods, and current statistics are solid. Nonetheless, the comments below should be appropriately answered and presented in the revised manuscript.

- It seems that there are two distinctive clusters for Narok in Fig 3A. Any possible explanation?

- Retention time/mz is not necessary for Fig 3B.

- Table 1 is duplicated in Result and Material & Method.

- What is the rationale that the authors performed OPLS-DA for Machakos and Nanyuki? Same quadrat location and particularly relative closeness do not sound reasonable. For instance, the relative distance between Nanyuki and Narok are close and it is even closer from HCA.

- Similarly, it seems that there are two distinctive clusters for Machakos in Fig 4A. Any possible explanation? Were they collected from different origin?

- For OPLS-DA, the authors should provide score plot, k-fold cross-validation and random permutation with R2 and Q2 values.

- In Fig 5, x-axis indicates relative abundance (mg). Did the authors perform absolute quantification?

- The authors are requested to provide the information morphological characters (e.g. pictures).

- Present the score plot of OPLS-DA for the five groups. How did the authors obtain the list of the metabolites in Table 3?

- Where is 3.3 part?

- Provide detailed information on data processing of LC-MS, particularly peak identification.

- The authors should discuss some putative linkage between the metabolite profiles and geo-environmental condition (e.g. temperature, total rain fall)

Author Response

RESPONSE TO REVIEWER 2 COMMENTS

Point 1: It seems that there are two distinctive clusters for Narok in Fig 3A. Any possible explanation?

Response 1: The sub clustering may be due to intraspecific hybridization, with members from populations occurring in adjacent ecological zones. This information has been added in the manuscript

Point 2- Table 1 is duplicated in Result and Material & Method.

Response 2- Table 1 in the Material and Method has been deleted

Point 3:  What is the rationale that the authors performed OPLS-DA for Machakos and Nanyuki? Same quadrat location and particularly relative closeness do not sound reasonable. For instance, the relative distance between Nanyuki and Narok are close and it is even closer from HCA.

Response 3: Similarity in terms of metabolites production between the two populations was an additional factor that informed the performance of OPLS-DA

Point 4:  Similarly, it seems that there are two distinctive clusters for Machakos in Fig 4A. Any possible explanation? Were they collected from different origin?

Response 4: This sub clustering may be due to intraspecific hybridization, with members from populations occurring in adjacent ecological zones

Point 5: For OPLS-DA, the authors should provide score plot, k-fold cross-validation and random permutation with R2 and Q2 values.

Response 5: Both k-fold cross-validation and random permutation with R2 and Q2 values has been provided in the supplementary material

Point 6:  In Fig 5, x-axis indicates relative abundance (mg). Did the authors perform absolute quantification?

Response 6: Yes, absolute quantification was performed

Point 7: The authors are requested to provide the information morphological characters (e.g. pictures).

Response 7: The morphological characters distinguishing the two varieties have been explained further in the introduction

Point 8: Present the score plot of OPLS-DA for the five groups. How did the authors obtain the list of the metabolites in Table 3?

Response 8: The list of the metabolites in table 3 were obtained from Figure 3B as the discriminants of the five populations

Point 9: Where is 3.3 part?

Response 9: The typographic error showing 3.2 followed by 3.4 has been rectified

Point 10:  Provide detailed information on data processing of LC-MS, particularly peak identification.

Response 10: Details on MSE fragments, UV and ppm error of the identified compounds have been added to table 2

Point 11: The authors should discuss some putative linkage between the metabolite profiles and geo-environmental condition (e.g. temperature, total rain fall)

Response11: putative linkage between the metabolite profiles and geo-environmental condition has been discussed especially on the coastal (Gazi) population

Reviewer 3 Report

This is a descriptive manuscript devoted to the determination of chemical content of different Dodonaea viscosa species, growing on the mainland (D. viscosa var angustifolia) and on the coast (D. viscosa var viscosa). The main idea of the manuscript is to classify different D. viscosa plants found in Kenya using metabolomic composition of these species and to find possible chemical markers that may help in the botanical classification of plants. The most important findings of the work are: 1) four flavonoid and diterpenoid compounds with antimicrobial activity were isolated from coastal plants; 2) the metabolomic profiles of different D. viscosa plants are influenced by geo-environmental conditions; 3); the discriminant compounds were identified in plants for each geographical region; 4) the current morphological classification of D. viscosa plants in Kenya was not supported by the metabolomic analysis performed in the study.

The manuscript is written in clear, concise style, nevertheless, there are several inconsistent sentences and typos. The experimental design is well thought out, but some parts lack more detailed description, and I have critical comments and questions concerning the obtained results:

  1. The PCA score plot (Figure 3A) contains five groups of samples referring to 5 regions. In the Experimental section (line 267) it is mentioned, that only three (3) samples from individual plant were studied for each region. Firstly, three samples is the absolute minimum for the statistical analysis and are inappropriate for publishing in highly rated journal. Secondly, as it is shown on Figure 3A, there are more than three samples for each group. It should be clearly noted, whether these are replicates of each three samples, or different ones. The latter should be noted in the Experimental section.
  2. Table 2 contains 26 compounds that were reported for the first time in viscosa. But I haven’t found in the manuscript, even in Supplementary, LC-MS data, confirming that all these compounds were identified in the extracts. LC-MS data should be added not only for four isolated compounds, but for the extracts also. (Moreover, it is doubtful, that it is possible to correlate chemical formula and exact compound without a standard). The m/z error should be added to the table for each compound, or theoretical [M-H]- value.
  3. Table 3 should contain the regional assignment, otherwise it is impossible to distinguish, which compounds correspond to each region, and compound should be sorted inside each region according to the chosen parameter (RT, m/z, etc.).     
  4. All tables should be sorted according to the retention time, or m/z, or other parameters, otherwise the tables filling logic is not clear.
  5. May be I missed somewhere, but why the exact four compound (hautriwaic acid, its lactone, catechin and tetrahydroxy-trimethoxyflavone) were chosen for the purification? Solvents for the extraction of metabolites are different in 2.2 and 2.5 of Experimental section. Explain, please, this difference.
  6. What was the purpose of testing the antimicrobial activity of compounds that are known to have precise antimicrobial activity (e.g. catechin)?
  7. How did you define and distinguish viscosa var angustifolia and D. viscosa var viscosa during sample collection? Only depending on the region, or there are other morphological factors? Is it possible that D. viscosa var angustifolia spread to the North of the mainland?
  8. In the Experimental section (lines 306-308) the fragmentation parameters are mentioned, but the fragmentation patterns for the compound are shown neither in the manuscript, nor in Supplementary material.
  9. Line 23 - viscosa var angustifolia is better here
  10. Lines 28-29 – replace “five” with “four” or add “including” one coastal
  11. Line 36 – “Fifteen” discriminant compouns are mentioned in the Abstract, while “fourteen” are written in the main body (line 125) and only “eleven” are assigned on Figure 3B. Define, please, the correct number of compounds.
  12. Line 99 – “Table 1”
  13. Figure 2 should contain full names of compounds on the right had side of the heat map.
  14. Line 131 – replace Figure 3A. with “Figure 3. A)”
  15. Line 136 - replace Figure 4A. with “Figure 3. A)”
  16. Line 165 – replace compounds belong with “compound belongs”
  17. Figure 5 should contain full names of compounds on the left had side of the plot
  18. Figure 6 – the atom labels on the hautriwaic acid structure should be enlarged as for the other structures
  19. Lines 247-249 – The whole sentence should be rewritten in the clearer way
  20. Line 283 – the duplicate table should be removed
  21. Supplementary 1.1 – please, specify the m/z values more precisely
  22. Figure S6 – this is more like a chemical noise with an intensity of about 1000-3000
  23. Figures S12, S18and S24 should be zoomed to be able to see the isotopic distribution of each compound. Fragment spectra with assignment should also be shown for the purified compound structure confirmation

Author Response

RESPONSE TO REVIEWER 3 COMMENTS

Point 1: The PCA score plot (Figure 3A) contains five groups of samples referring to 5 regions. In the Experimental section (line 267) it is mentioned, that only three (3) samples from individual plant were studied for each region. Firstly, three samples is the absolute minimum for the statistical analysis and are inappropriate for publishing in highly rated journal. Secondly, as it is shown on Figure 3A, there are more than three samples for each group. It should be clearly noted, whether these are replicates of each three samples, or different ones. The latter should be noted in the Experimental section

Response 1: For each biological replicate of the materials collected, there were three technical      replicates analyzed and thus the increase in the number of samples for each group.

Point 2: Table 2 contains 26 compounds that were reported for the first time in viscosa. But I haven’t found in the manuscript, even in Supplementary, LC-MS data, confirming that all these compounds were identified in the extracts. LC-MS data should be added not only for four isolated compounds, but for the extracts also. (Moreover, it is doubtful, that it is possible to correlate chemical formula and exact compound without a standard). The m/z error should be added to the table for each compound, or theoretical [M-H]- value.

Response 2: The MSE fragments, ppm error and UV data of the reported compounds has been added to the table 2

Point 3: Table 3 should contain the regional assignment, otherwise it is impossible to distinguish, which compounds correspond to each region, and compound should be sorted inside each region according to the chosen parameter (RT, m/z, etc.).

Response 3: The table has been enriched with regional assignment and sorted in terms of RT    

Point 4: All tables should be sorted according to the retention time, or m/z, or other parameters, otherwise the tables filling logic is not clear.

Response 4:Table 2 has been sorted based on retention time

Point 5 - May be I missed somewhere, but why the exact four compound (hautriwaic acid, its lactone, catechin and tetrahydroxy-trimethoxyflavone) were chosen for the purification? Solvents for the extraction of metabolites are different in 2.2 and 2.5 of Experimental section. Explain, please, this difference.

Response 5- The four compounds were not chosen for purification but were rather obtained after subjecting the leaf extract from the coastal collection to column chromatography. The solvents in 2.2 and 2.5 are different because the extraction for isolation requires both polar and non polar solvents unlike for LC/MS analysis

 Point 6- What was the purpose of testing the antimicrobial activity of compounds that are      known to have precise antimicrobial activity (e.g. catechin)?

Response 6- The purpose of testing the antimicrobial activity for the isolated compounds was to compare their activity with that of the extracts from the five populations of D. viscosa (antimicrobial activity of the extracts added).

Point 7- How did you define and distinguish viscosa var angustifolia and D. viscosa var viscosa during sample collection? Only depending on the region, or there are other morphological factors? Is it possible that D. viscosa varangustifolia spread to the North of the mainland?

Response 7- The morphological differences between the two varieties of D. viscosa that informed the identification during sample collection have been added in the introduction part.

D. viscosa var. viscosa, a thick bush/shrub that grows to 3-4 m with a fruit capsule that is usually white or brown with 2 large wings and bisexual flowers. The leaves are somewhat larger than D. viscosa var angustifolia. This variant is referred to locally as Mkaa pwani (Swahili) and is more confined to the coastal parts of Kenya while D. viscosa var. angustifolia (L.f.) Benth.is a shrub or small tree 1-6 m with pinkish or reddish 2-3 winged fruits and unisexual yellow- green flowers with a much wider distribution occurring inland in Kenya’

Point 8- In the Experimental section (lines 306-308) the fragmentation parameters are mentioned, but the fragmentation patterns for the compound are shown neither in the manuscript, nor in Supplementary material.

Response 8- Details of the fragmentation parameters for all the identified compounds have been added to table 2

Point 9- Line 23 - viscosa var angustifolia is better here

Response 9- D. viscosa var angustifolia has been inserted

Point 10- Lines 28-29 – replace “five” with “four” or add “including” one coastal

Response 10- The sentence has been edited  to read; D. viscosa var. angustifolia populations collected from four different geographical regions (Nanyuki, Machakos, Nairobi, Narok) and one coastal D. viscosa var viscosa (the Gazi)

Point 11- Line 36 – “Fifteen” discriminant compounds are mentioned in the Abstract, while “fourteen” are written in the main body (line 125) and only “eleven” are assigned on Figure 3B. Define, please, the correct number of compounds.

Response 11- The correct number of discriminant compounds is eleven. The correction has been factored to reflect this in both the abstract and the main body.

Point 12- Line 99 – “Table 1”

Response 12- Table 1 corrected

Point 13- Figure 2 should contain full names of compounds on the right had side of the heat map.

Response 13- Full names of the compounds have been written on the right had side of the heat map.

Point 14- Line 131 – replace Figure 3A. with “Figure 3. A)”

Response 14- Figure 3A replaced with “Figure 3. A

  Point 15- Line 136 - replace Figure 4A. with “Figure 4. A)”

Response 15- Figure 4A replaced with “Figure 4. A

Point 16- Line 165 – replace compounds belong with “compound belongs”

 Response 16- compounds belong replaced with “compound belongs

Point 17- Figure 5 should contain full names of compounds on the left had side of the plot

Response 17- full names of compounds have been written on the left had side of the plot in figure 5

Point 18- Figure 6 – the atom labels on the hautriwaic acid structure should be enlarged as for the other structures

Response 18- the atom labels for hautriwaic acid in Figure 6 have been enlarged

Point 19- Lines 247-249 – The whole sentence should be rewritten in the clearer way

Response 19- The sentence has been rewritten

Point 20- Line 283 – the duplicate table should be removed

Response 20- The duplicate table has been removed

Point 21- Supplementary 1.1 – please, specify the m/z values more precisely

Response 21- the m/z values specified to four decimal places

Point 22- Figure S6 – this is more like a chemical noise with an intensity of about 1000-3000

Response 22-The correct figure has been inserted

Point 23- Figures S12, S18and S24 should be zoomed to be able to see the isotopic distribution of each compound.

Response 23- Figures S6, S12, S18and S24 have been zoomed

Round 2

Reviewer 2 Report

All comments were appropriately answered and accordingly reflected in the revised mansucrpt.

Reviewer 3 Report

Thank you for the detailed responses and corrections of text and figures.